# circSLC41A1 Resists Porcine Granulosa Cell Apoptosis and Follicular Atresia by Promoting SRSF1 through miR-9820-5p Sponging

**DOI:** 10.3390/ijms23031509

**Published:** 2022-01-28

**Authors:** Huiming Wang, Yi Zhang, Jinbi Zhang, Xing Du, Qifa Li, Zengxiang Pan

**Affiliations:** 1College of Animal Science and Technology, Nanjing Agriculture University, Nanjing 210095, China; 2019105085@njau.edu.cn (H.W.); 2020105023@stu.njau.edu.cn (Y.Z.); duxing@njau.edu.cn (X.D.); liqifa@njau.edu.cn (Q.L.); 2College of Animal Science and Food Engineering, Jinling Institute of Technology, Nanjing 211169, China; zhangjinbi@jit.edu.cn

**Keywords:** circRNA, non-coding RNA, ceRNA, circSLC41A1, miR-9820-5p, SRSF1, follicular atresia, granulosa cell apoptosis

## Abstract

Ovarian granulosa cell (GC) apoptosis is the major cause of follicular atresia. Regulation of non-coding RNAs (ncRNAs) was proved to be involved in regulatory mechanisms of GC apoptosis. circRNAs have been recognized to play important roles in cellular activity. However, the regulatory network of circRNAs in follicular atresia has not been fully validated. In this study, we report a new circRNA, circSLC41A1, which has higher expression in healthy follicles compared to atretic follicles, and confirm its circular structure using RNase R treatment. The resistant function of circSLC41A1 during GC apoptosis was detected by si-RNA transfection and the competitive binding of miR-9820-5p by circSLC41A1 and SRSF1 was detected with a dual-luciferase reporter assay and co-transfection of their inhibitors or siRNA. Additionally, we predicted the protein-coding potential of circSLC41A1 and analyzed the structure of circSLC41A1-134aa. Our study revealed that circSLC41A1 enhanced SRSF1 expression through competitive binding of miR-9820-5p and demonstrated a circSLC41A1–miR-9820-5p–SRSF1 regulatory axis in follicular GC apoptosis. The study adds to knowledge of the post-transcriptional regulation of follicular atresia and provides insight into the protein-coding function of circRNA.

## 1. Introduction

Follicular atresia is a natural physiological phenomenon observed in all stages during follicular development in vertebrates [1]. In pigs, which are not only important for pork production but are also recognized as large animal models for biomedical research [2], the primordial follicle reservoir contains approximately 5 million primordial follicles at puberty. However, a large number of follicles undergo atresia and disappear before reaching 6 mm in diameter, and result in less than 14% for ovulation [3]. As a major component of the follicle, granulosa cells (GCs) function by closely interacting with the oocyte and theca cells and producing paracrine substances. Research has demonstrated that GCs’ proliferation and function are essential in follicle development and maturation, while the GC apoptosis rate increases significantly with the progression of follicular atresia [4].

Non-coding RNAs (ncRNAs) have long attracted attention for their essential roles in post-transcriptional gene regulation. Among ncRNAs, microRNAs (miRNAs) are a group of the most-studied small ncRNAs ~22 nt in length. The critical roles of miRNAs in GC apoptosis and the follicular atresia process have been well studied and proved in humans, mice, pigs, and bovines [5]. In addition, circular RNA (circRNA) is a new type of ncRNA produced from precursor mRNA through back-splicing of precursor RNA. circRNAs are located in the cellular nucleus or cytoplasm and regulate gene expression through transcriptional or post-transcriptional pathways. circRNAs are abundantly expressed in the blood [6], saliva [7], and exosomes [8] due to their stable circular structure, and thus may serve as potential biomarkers. The most famous circRNA mechanisms are the circRNA-associated-ceRNA (competing endogenous RNA) networks [9]. In this case, circRNAs competitively bind to miRNA to regulate the expression level of the target gene [10]. There has also been evidence for circRNA translation of functional protein products [11]. Recent studies have revealed the critical roles of circRNAs in GC proliferation and apoptosis during mouse [12], bovine [13,14], and goat [15] follicular development, and ovarian senescence [16]. Our previous study has shown an abundant distribution and significant shift of miRNAs [17] and circRNAs [18] in porcine antral follicles during atresia. A detailed separation of granulosa, theca, and cumulus cells further proved the role of aberrantly expressed circRNAs in porcine antral follicular atresia [19]. The differential expression profiles of miRNA and circRNAs in healthy follicles (HFs) and atretic follicles (AFs) suggested a possible circRNA-miRNA regulatory network during atresia. More specifically, the circRNA sequencing highlighted a new circRNA, circSLC41A1, which is spliced from the solute carrier family 41-member one gene (*SLC41A1*) and is more highly expressed in HFs compared to AFs. The preliminary bioinformatic analysis implied its potential as a ceRNA in *SRSF1* post-transcriptional regulation.

*SRSF1* (SF2/ASF) is a prototypical serine-/arginine-rich (SR) protein that is essential for cell viability. It is involved in splicing site recognition at intron-exon junctions, exon skipping, and regulation of alternative splicing [20,21]. *SRSF1* can also bind to ribonucleoproteins and participate in stabilization, transportation, intracellular localization, and mRNA translation [22,23,24]. Thus, loss or inactivation of *SRSF1* results in genomic instability, G2 cell cycle arrest, and apoptosis because many related factors are expressed via alternative splicing [25]. It has been reported that miRNA is involved in *SRSF1* regulation. For example, miR-505-3p targets *SRSF1* to inhibit the expression of puberty-related genes, thus regulating sexual maturity [26]; also, both miR-28 and miR-505 target *SRSF1* to control mouse embryonic fibroblasts (MEF) proliferation and survival [27]. Additionally, SRSF1 was reported to bind to circRNA [28] and lncRNA [29], as RNA-binding proteins (RBPs) in glioma cells and trophoblast cells, respectively, to regulate the cell proliferation process. However, *SRSF1* function and its regulatory mechanism are rarely studied in reproduction, especially during GC apoptosis and follicular atresia. Interestingly, our previous RNA-seq study highlighted a higher expression of *SRSF1* in HFs than AFs, which hinted at its regulatory potential in GC apoptosis [4].

In this study, we reported that a newly identified circRNA, circSLC41A1, guaranteed the expression of SRSF1 by sponging miR-9820-5p, thus further resisting GC apoptosis and follicular atresia. The possible translation product of circSLC41A1 was also analyzed. Our study demonstrated a circSLC41A1-miR-9820-5p-SRSF1 regulatory axis in GC apoptosis and predicted that circSLC41A1 has the potential to encode small peptides with antioxidant functions, which not only adds new knowledge about SRSF1 function in apoptosis and the ovarian physiological process but also provides new ideas for further studies on the function of ovarian circRNA. These results provided a basis for exploring the molecular mechanism underlying follicular atresia and serve as references for animal reproduction and reproductive medicine.

## 2. Results

### 2.1. Identification and Validation of circSLC41A1

According to our previous circRNA sequencing and bioinformatic analysis, circSLC41A1 is a 1027 bp circular RNA generated by back splicing of *SLC41A1* gene exon 2 and is decreased during follicular atresia [3]. The circular structure of circSLC41A1 was verified by amplification with specific divergent primers followed by Sanger sequencing. The result confirmed that circSLC41A1 was reverse spliced from the exon 2 regions of the *SLC41A1* mRNA precursor (Figure 1A). The stability of circSLC41A1 was further verified by qPCR amplification with or without RNase R digestion. We observed significant reduced liner *SLC41A1* and increased circSLC41A1 levels after digestion, demonstrating that circSLC41A1 was resistant to RNase R and structurally stable (Figure 1B). Moreover, a significant decrease of circSLC41A1 was detected in AFs relative to HFs via qRT-PCR (Figure 1C). To demonstrate circSLC41A1 cellular localization, we conducted a FISH analysis. The junction probe detected abundant cytoplasmic circSLC41A1 expression in GCs (Figure 1D). These results revealed the existence and circular structure of circSLC41A1, which might play a role in porcine follicular atresia and GC apoptosis.

### 2.2. circSLC41A1 Inhibits GCs Apoptosis

To explore circSLC41A1 function in porcine GCs, a specific small interfering RNA, si-circSLC41A1, was designed across the splicing site. The transfection of si-circSLC41A1 interfered with circSLC41A1 expression level after 24 h but not its corresponding linear mRNA in GCs (Figure 2A). The GC apoptosis rate was detected by Annexin V-FITC/PI staining assay, and the level of apoptosis marker protein, C-CASP3 was detected by WB after 72 h of transfection. The results showed that both the GC apoptosis rate (Figure 2B) and C-CASP3 levels (Figure 2C) increased significantly. These results suggested that circSLC41A1 had a positive effect on GC survival and inhibited GC apoptosis.

### 2.3. circSLC41A1 Is a Possible Sponge for miR-9820-5p

According to our previous miRNA-seq result and bioinformatic prediction, miR-9820-5p was highlighted as its expression level is raised during follicular atresia and may bind to circSLC41A1. The expression of miR-9820-5p in HFs and AFs was verified by qRT-PCR, and the expressed level was significantly higher in AFs (Figure 3A). We also observed a negative correlation between circSLC41A1 and miR-9820-5p expression levels in individual follicles (Figure 3B). To demonstrate the biological function of miR-9820-5p in GCs, its mimics or inhibitors were transfected to GCs effectively (Appendix A). The FACS results suggested that the GC apoptosis rate increased after miR-9820-5p mimics treatment (Figure 3C) and decreased significantly after its inhibitor transfection (Figure 3D), implying a pro-apoptotic effect of miR-9820-5p in GCs. In addition, dual-luciferase activity assays were applied to verify the binding sites of the miR-9820-5p on circSLC41A1. The results suggested that miR-9820-5p mimics bond to the circSLC41A1 wild-type vector (WT) but not the mutant vector (MUT) (Figure 3E). The cytoplasmatic co-localization ofcircSLC41A1 and miR-9820-5p was further detected by FISH assay in GCs (Figure 3F). The above results suggested that circSLC41A1 was a possible miR-9820-5p sponge in GCs.

### 2.4. SRSF1 Resists GC Apoptosis and Follicular Atresia

According to our previous RNA-seq result and interaction network prediction, *SRSF1* was screened as a possible downstream effector gene of the circSLC41A1-miR-9820-5p axis (Figure 4A). The decreased expression levels of *SRSF1* during follicular atresia were verified by qRT-PCR (Figure 4B). Then, an effective small interfering RNA for *SRSF1*, termed si-SRSF1, was designed to knock down *SRSF1* (Figure 4C,D), and FACS results showed that the GC apoptosis rate increased significantly after knockdown of *SRSF1* (Figure 4E).

### 2.5. MiR-9820-5p Promotes GC Apoptosis by Targeting SRSF1

To further explore the interaction between miR-9820-5p and *SRSF1*, the binding sites of miR-9820-5p and SRSF1 were predicted by RNAhybrid (Figure 5A), then dual-luciferase activity assays were applied to verify their binding. The results suggested that miR-9820-5p could directly bond to wild-type *SRSF1* (SRSF1-WT) but not the mutant type (SRSF1-MUT) (Figure 5B). Additionally, after transfection of miR-9820-5p mimics into GCs, both mRNA and protein levels of *SRSF1* were significantly reduced (Figure 5C,D), while transfection of miR-9820-5p inhibitors showed an opposite SRSF1 expression trend (Figure 5E,F). Moreover, co-transfection of miR-9820-5p inhibitors and si-SRSF1 revealed that si-SRSF1 could reverse the GC apoptosis rate change caused by miR-9820-5p inhibition via flow cytometry (Figure 5G). These investigations indicated that miR-9820-5p promotes porcine GC apoptosis by targeting *SRSF1*.

### 2.6. circSLC41A1 Inhibits GC Apoptosis by Upregulating SRSF1 through Sponging miR-9820-5p

To explore whether circSLC41A1 regulates *SRSF1* and GC apoptosis through sponging miR-9820-5p, we co-transfected si-circSLC41A1 and miR-9820-5p inhibitors into GCs and detected the changes in *SRSF1* expression levels and the GC apoptosis rate. The results suggested that the decreased *SRSF1* mRNA and protein levels caused by si-circSLC41A1 were reversed by an additional miR-9820-5p inhibitor (Figure 6A,B). In addition, the increased GC apoptosis due to si-circSLC41A1 was also reversed by an additional miR-9820-5p inhibitor via flow cytometry (Figure 6C). These results confirmed our hypothesis that circSLC41A1 plays a role as an miR-9820-5p sponge to upregulate *SRSF1* and thus resist GC apoptosis during follicular atresia.

### 2.7. Evaluation of the Coding Ability of circSLC41A1

Some exonic circRNAs have been reported to encode small peptides with biological functions. We analyzed the putative open reading frame (ORF), internal ribosome entry site (IRES), and RRACH motif, which contains the possible N6-methyladenosine (m^6^A) modification site, in circSLC41A1. As shown in Figure 7A, we found four potential spanning junction ORFs, which were termed ORF1–4, that encoded 134, 102, 115, and 105 aa peptides, respectively, in circSLC41A1. As IRES is required for 5ʹ cap-independent translation [30], we predicted IRES sites at +188~201 bp and +631~643 bp, termed IRES1 and IRES2, respectively. The IRES2 was located close to the ORF1 initiation codon (ATG), which might be beneficial to promote ORF1 translation. In addition, a potential m^6^A site was predicted with high confidence at +698 bp, which is also close to the ORF1 initiation codon. According to the above results, we preliminarily speculated that the ORF1 of circSLC41A1 had the potential to encode small circSLC41A1-134aa peptides. Then, we compared circSLC41A1-134aa with the protein sequence encoded by SLC41A1 mRNA and found that the N-terminus of the peptide shares the same 125 aa sequence with SLC41A1 and that the 9 aa sequence (VFHTDSCDF) at the C-terminus was distinct. Conservation analysis showed that ORF1 and circSLC41A1-134aa are highly conserved among different species, implying that the ORF1 is translatable and 134aa plays an important biological function. (Appendix A). The secondary structure prediction suggested that 134aa might form relatively stable structures based on its α-helix as well its β-sheet and form active centers with unique biological functions based on its random coils (Appendix A). To further explore the 134aa function, the prediction of its physicochemical properties and transmembrane domain showed that the molecular formula was C_600_H_940_N_162_O_206_S_5_, that the molecular weight was 1913, that it was hydrophilic (Figure 7B), and that there was a transmembrane domain at the amino acids 103–125 (Figure 7C). Then the 134aa-bound ligand was predicted based on its tertiary structure (Figure 7D), which showed that the peptide might bind to ligands such as CA (Ca^2+^), NDP (NADPH), and 1AQ (Figure 7E) to participate in biological processes, including oxidation-reduction reactions and peptidoglycan catabolism. CA bind at amino acids 122 and 125 of 134aa, which happens to be its transmembrane domain. The 134aa may function as a transmembrane protein acting as an ion channel.

## 3. Discussion

In recent years, ncRNA became a hotspot of animal breeding and reproductive medicine. Characterized by a covalently closed-loop structure that is generated via reverse splicing, circRNA shows high stability and is widely distributed in human and animal tissues [31]. The high-throughput sequencing profiling of circRNA expression is a prerequisite for detailed explorations of mechanisms and functions of circRNAs during physiological and pathological processes [32]. However, it also left many unanswered questions regarding the detailed functionality of circRNA shifts. Our knowledge of how individual circRNA affects the physiological process of the follicle is still in infancy and needs further improvement and supplementation. Generally, the biological functions of circRNA involve competing with the pre-mRNA splicing of parental RNA [33], acting as ceRNAs that regulate target gene expression by efficient miRNA binding or sponging [34], and protein-binding and protein-coding abilities [35]. In the female ovary, circRNA profiles have been examined in humans [5], mouse [12], and domestic animals [18], which affirmed the research and application value of circRNA during follicular development, atresia, and disfunction. In this study, we identified a new SLC41A1-coded circRNA, circSLC41A1, which is more highly expressed in healthy follicles. We revealed that circSLC41A1 is associated with GC viability through the circSLC41A1-miR-9820-5p-*SRSF1* regulatory axis, extending our knowledge about the ncRNA regulatory network in ovarian GC apoptosis and follicular atresia (Figure 8).

Alternative splicing is a co-transcriptional process that is regulated by two highly conserved protein families, SR proteins and heterogeneous nuclear ribonucleoproteins (hnRNPs), which generally are active in or exhibit splicing processes. As one of the SR proteins, SRSF1 binds to splicing enhancers and activates the splicing of multiple gene transcripts [18]. SRSF1 was primarily identified as an oncogene that was frequently overexpressed in multiple types of human tumors. The reported regulatory pathways involves the regulation of apoptosis-related genes, such as BCLX, MCL1, and caspases 2 and 9 [36], activation of Wnt signaling by enhancing β-catenin biosynthesis [37] and promoting the expression of BIN1 and BIM isoforms lacking pro-apoptotic activity [25]. Our study, for the first time, demonstrated the biological function of *SRSF1* in ovarian follicles and proved its apoptosis-resistant effect in non-tumor tissue. In addition, the ceRNA hypothesis has proposed that mRNAs, lncRNAs, and circRNAs regulate each other through competition of the same miRNA binding, thus constructing a complex post-transcriptional regulatory network [38]. This was, to our knowledge, the first investigation showing that *SRSF1* expression was affected through a post-transcriptional regulation network in which circSLC41A1 competitively binds to miR-9820-5p to regulate the expression level of the target gene *SRSF1*, adding new insight into the subtle adjustment of *SRSF1* levels and activity.

There are two other possible effects of circSLC41A1 that are worthy of note. Firstly, although circRNA functions are generally irrelevant to its host gene, the product of circSLC41A1 would reduce the expression of liner mRNA and protein of SLC41A1. SLC41A1, which belongs to the solute carrier family, is a novel Mg^2+^ transporter that mediates Mg^2+^ efflux and maintains intracellular Na^+^ and Mg^2+^ homeostasis [39,40]. Studies have shown that SLC41A1 caused apoptosis by inhibiting Mg^2+^-dependent AKT-mTOR signal transmission and activating BAX expression in pancreatic tumor cells [41], which agrees with our results that circSLC41A1 levels dropped during atresia and the fact that it resisted GC apoptosis. However, whether competitive splicing of circSLC41A1 affects SLC41A1 in apoptosis awaits further investigation.

Secondly, circSLC41A1 showed a high protein-coding potential. The translation of endogenous circular RNAs was first reported in 2017. It was shown that circ-SHPRH produced a novel tumor suppressor protein, SHPRH-146aa. The protein worked in synergy with the full-length SHPRH protein and served as a protective decoy molecule to decrease its degradation [42]. In our study, the bioinformatic analysis not only suggested the coding ability of circSLC41A1 ORF1 but also revealed that the small peptide circSLC41A1-134aa was also a transmembrane protein, which largely overlaps with the SLC41A1 protein and with specific Ca^2+^-binding potential. Based on this evidence, we supposed that circSLC41A1-134aa perhaps functions as an assistant protein of SLC41A1 to directly transport, or simply attract Mg^2+^ and Ca^2+^, to maintain the plasma membrane [43]. In addition, based on the functional annotation of its tertiary structure, circSLC41A1-134aa might participate in oxidoreductase activity, N-acetylmuramyl-L alanine amidase activity, and peroxisomes, which play a crucial role in the regulation of the antioxidant system. Thus, circSLC41A1 may reduce GC apoptosis and inhibit follicular atresia through balancing SLC41A1 and circSLC41A1-134aa products.

In summary, our results showed that a newly identified circSLC41A1 downregulated during porcine follicular atresia. It inhibited porcine GC apoptosis through competitive binding of miR-9820-5p with *SRSF1*, which provided a new post-transcriptional axis for *SRSF1* regulation and insights into the mechanistic exploration of follicular atresia. The high coding potential of circSLC41A1 had also suggested a possible co-transport function of circSLC41A1-134aa with SLC41A1 itself. Detailed understanding of circSLC41A1-coded small peptides in follicular atresia requires subsequent study.

## 4. Materials and Methods

### 4.1. Follicle Collection and Isolation

All experimental procedures were performed according to the guidelines of the Administration of Animal Care and Use and were approved by the Animal Ethics Committee of Nanjing Agricultural University, Nanjing, Jiangsu, China (SYXK2011-0036, 6 December 2011). The porcine ovaries were obtained from healthy, unstimulated commercial gilts (Duroc–Landrace–Yorkshire, 200-day-old, average weight 90 kg) after routine slaughtering procedures from Sushi slaughterhouse in Huai’an, Jiangsu, and brought back to the laboratory as soon as possible in normal saline with 1% penicillin-streptomycin. The ovarian tissue was washed and cut with a scalpel in a petri dish containing PBS. Antral follicles with a diameter of 3–5 mm were isolated using small scissors and forceps [44]. Follicles were divided into healthy and early atresia according to a uniform determination based on follicle appearance, antral GC density, and progesterone (P4)/estradiol (E2) ratios [45]. Briefly, ovarian follicles with a clear and transparent appearance, visible small blood vessels, low GC density (≤25,000/μL), and a low P4/E2 ratio (≤1.5) were categorized as HFs. Ovarian follicles with a turbid orange appearance, visible large blood vessels, high GC density (≥25,000/μL), and a high P4/E2 ratio (≥1.5) were categorized as AFs. According to standards, 8 healthy and 8 atretic follicles were selected for qRT-PCR.

### 4.2. Cell Culture and Transfection

The porcine ovaries were washed with 75% alcohol and sterile physiological saline in turn. The GCs were obtained from clear antral follicles using a syringe with a 20-gauge needle and then cultured at 37 °C in 5% CO2 using DMEM/F-12 medium (Gibco, Carlsbad, CA, USA) containing 10% fetal bovine serum (FBS) (Gibco, Carlsbad, CA, USA) and 1% penicillin-streptomycin in vitro. Generally, 5 ovaries were used to cultivate GCs in a 12-well plate, and 7.5 ovaries were used to cultivate GCs in a 6-well plate. The final volume in the 12-well culture plate was 1 mL/well and the cell concentration was 0.8 × 10^6^ live cells/well, while the final volume in the 6-well culture plate was 2 mL/well and the cell concentration was 1.65 × 10^6^ live cells/well. HEK 293 cells were incubated at the same condition in a DMEM medium (Sigma, St. Louis, MO, USA). For the FISH assay, the GCs were cultured on coverslips placed in 6-well plates. Transfection was applied after 36 h of culture, when cells reached 50–80% confluence, using LipofecctamineTM 3000 (Invitrogen, Carlsbad, CA, USA) and Opti-MEM (Gibco, Carlsbad, CA, USA) according to the manufacturer’s instructions. Inhibitors and mimics of circRNA siRNA and miRNA were synthesized by GenePharma (Shanghai, China) (Appendix A).

### 4.3. RNA Preparation and qRT-PCR

Total RNA was extracted after 24 h of siRNA treatment or 48 h of miRNA mimic and inhibitor treatments using Trizol reagent (Vazyme, Nanjing, Jiangsu, China), following the manufacturer’s instructions, and qualified by NanoDrop and agarose electrophoresis. RNA samples with OD260/OD280 between 1.7 and 2.0 and showing clear 18 s and 28 s bands were used for the first-strand cDNA synthesis with a PrimeScript RT Master Mix (Vazyme, Nanjing, Jiangsu, China) kit and then stored at −20 °C. The 10 uL RT system included 500 ng RNA, 2 uL gDNA wiper Mix, ddH_2_O (up to 6 uL), and 2 uL 5 × HiScript III qRT SuperMix. Quantitative determination of circRNA and miRNAs were applied by qPCR (ABI StepOne system, Applied Biosystems, Carlsbad, CA, USA) using an AceQ qPCR SYBR Green Master Mix Kit (Vazyme, Nanjing, Jiangsu, China), according to the manufacturer’s instructions. The 10 μL qPCR system included 1 μL cDNA, 5 μL AceQ qPCR SYBR Green Master Mix, 3.4 μL ddH_2_O, 0.2 μL primer 1, 0.2 μL primer 2, and 0.2 μL ROX Reference Dye 2. GAPDH was used as an endogenous control for circSLC41A1 and coding genes. To measure miR-9820-5p expression levels, miR-9820-5p stem-loop primer was designed to synthesize cDNA, and the specific primers of miR-9820-5p were designed for qRT-PCR. U6 was used as the internal control for miRNA. The relevant primer information is shown in Appendix A.

### 4.4. RNase R Digestion and RT-PCR

The extracted RNA of each GC sample was divided into two equal aliquots. One aliquot was directly subjected to reverse transcription, while the other was treated by RNase R (Geneseed, Guangzhou, Guangdong, China) before reverse transcription. The 20 μL digestive system included 5 μg RNA, 2 μL 10 × Reaction Buffer, 15 U RNase R (20 U/μL), and RNase-Free water. The reaction was performed at 37 °C for 15 min and then at 70 °C for 10 min. PCR was then performed using a 2 × Vazyme Lamp Master Mix kit (Vazyme, Jiangsu, Nanjing, China), according to the instructions. Briefly, under the reaction of 2 × Taq Master Mix, cDNA, and primers, the gene sequence was amplified under a certain temperature program. Then, PCR products were verified by agarose gel electrophoresis and Sanger sequencing (Bioengineering, Shanghai, China).

### 4.5. Dual-Luciferase Activity Assay

The circSLC41A1 sequence containing wild-type (WT) or mutant-type (Mut) binding sites for miR-214-5p were inserted into pmirGLO Dual-Luciferase miRNA Target Expression Vector (Promega, Madison, WI, USA). The pmirGLO vector containing the circRNA sequence of the binding site and miR-214-5p mimics was synthesized by Qingke Biotechnology Co, Nanjing, China. The renilla and firefly luciferase activities were assessed using the Dual-Luciferase Assay kit (Promega, Madison, WI, USA) with a luminescence detector (Promega Corporation, Madison, WI, USA). Briefly, after GCs were lysed with Passive Lysis Buffer (1×), the cell sample was first reacted with LAR2, then the reaction was terminated by STOP (1×). The binding site and mutation site sequences are shown in Appendix A.

### 4.6. Apoptosis Assay

After 72 h of treatment, GC apoptosis was measured with an Annexin V-fluorescein isothiocyanate (V-FITC)/propidium iodide (PI) staining kit (Vazyme, Nanjing, Jiangsu, China) according to the manufacturer’s protocol. Briefly, GCs were stained with V-FITC/PI. The percentage of apoptotic cells was detected by flow cytometry (Becton Dickinson FACS Calibur, Beckman Coulter, Brea, CA, USA) based on the principle of fluorescence-activated cell sorting (FACS). The apoptosis rate was analyzed using FlowJo v7.6 software (Stanford University, Stanford, CA, USA) and calculated using the following equation: percent of cells in the right upper quadrant + percent of cells in the right lower quadrant.

### 4.7. Western Blotting

Porcine GCs were cultured in 6-well plates. RIPA buffer containing 1% PMSF (Bioworld, Nanjing, Jiangsu, China) was added to lyse the cells after 72 h treatment. A BCA determination kit (Beyotime, Shanghai, China) was applied to measure protein concentrations. Briefly, the BCA working solution and protein sample were mixed and reacted at 37 °C, then the complex was measured by Multiskan Go (Thermo, Boston, MA, USA) at 562 nm wavelength. Proteins (final concentration of 1–2 μg/uL,10–20 μg/lane) were separated on 15% SDS-polyacrylamide gels and transferred to nitrocellulose membranes. The antibodies used in this study were polyclonal anti-SRSF1 (1:1000 dilution, 12929-2-AP, Proteintech Group, Rosemont, IL, USA), polyclonal anti-tubulin (1:1000 dilution, #6181S, Cell Signaling Technology, Boston, MA, USA), polyclonal anti-Cleaved Caspase-3 (C-CASP3) (1:1000 dilution, #9664S, Cell Signaling Technology, Boston, MA, USA), and the secondary antibody (1:2000 dilution, SA00E1–2, Proteintech Group, Rosemont, IL, USA). Protein levels were detected by an imaging system (Image LAS-4000, Bio-Rad, Hercules, CA, USA) with ECL Plus reagent (Promega, Madison, WI, USA) and analyzed using ImageJ. The results were normalized with Tubulin. Each experiment was performed three times.

### 4.8. Fluorescence In Situ Hybridization (FISH)

GCs were cultured on coverslips, fixed in 4% paraformaldehyde (containing DEPC) for 20 min, washed while shaken with PBS (pH 7.4) three times, permeabilized with 0.1% Triton X-100, and proteinase K (20 µg/mL) was finally added for 5 min for digestion [18]. The FAM-labelled probes were specifically synthesized for circSLC41A1 and miR-9820-5p, respectively (Servicebio technology, Wuhan, Hubei, China), and DAPI was used to stain the cell nuclei. All procedures were conducted according to the manufacturer’s instructions (Sevicebio, Wuhan, Hubei, China), including probe denaturation, denaturation, hybridization, elution, hybridization signal amplification, mounting, and probe chromogenic detection. The fluorescent images were finally acquired using a Nikon upright fluorescence microscope (Nikon DS-U3, Tochigi, Japan). Each experiment was performed three times. The probe sequences are shown in Appendix A.

### 4.9. Bioinformatics Analysis

The miRNA-gene interactions were predicted by PITA [46], Miranda [47], and TargetSpy [48] (https://arn.ugr.es/srnatoolbox/amirconstarget/) [49], according to the sequence. miRNA binding sites were predicted using RNAhybrid [50] (https://bibiserv.cebitec.uni-bielefeld.de/rnahybrid) [51]. Data integration was performed with intersection and VennDiagram [52] (http://bioinformatics.psb.ugent.be/webtools/Venn/) [53]. The IRES site, m^6^A site, and ORF were predicted, respectively, by IRESite [54] (http://www.iresite.org/) [55], SRAMP [56] (http://www.cuilab.cn/sramp) [57], and ORFfinder [58] (https://www.ncbi.nlm.nih.gov/orffinder/) [59]. The secondary structure, the tertiary structure, and the ligand-binding site of 135aa small peptide were predicted by the I-TASSER server [60] (http://zhang.bioinformatics.ku.edu/I-TASSER) [61]. The physical and chemical properties, hydrophobicity, and transmembrane domain of 134aa small peptide were predicted, respectively, by the ProtParam tool (https://web.expasy.org/protparam/) [62], ProtScale (https://web.expasy.org/protscale/) [63], and TMHMM-2.0 (https://services.healthtech.dtu.dk/service.php?TMHMM-2.0) [64].

### 4.10. Statistical Analysis

GraphPad Prism 6 (GraphPad Software, San Diego, CA, USA) was used to perform statistical analysis. Two-tailed Student’s *t*-tests were used to evaluate significance between the two groups. All data were presented as the means ± SEM of at least three independent experiments; *p* < 0.05 was considered statistically significant (*), and *p* < 0.01 considered a highly significant difference (**).

## Figures and Tables

**Figure 1 ijms-23-01509-f001:**
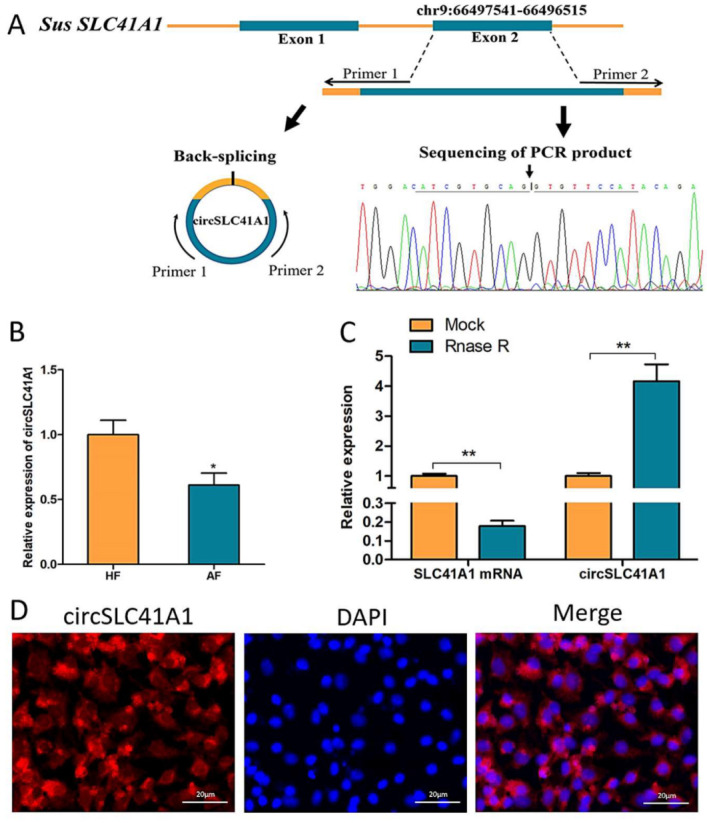
Identification and validation of circSLC41A1. (**A**) Sketch of the structure of circSLC41A1, which is generated from the *SLC41A1* gene exon 2 via back splicing. (**B**) *SLC41A1* mRNA and circSLC41A1 expression with and without RNase R digestion. Untreated GAPDH was used as an internal control. (**C**) Differential expression of circSLC41A1 in HFs and AFs detected by qRT-PCR (*n* = 8). (**D**) The cytoplasmic localization of *SLC41A1* in GCs detected by FISH. *SLC41A1* was labeled with red fluorescence, and the nuclei were stained by DAPI (blue). Scale bar: 20 μm. Data are expressed as the mean ± SEM, * *p* < 0.05, ** *p* < 0.01.

**Figure 2 ijms-23-01509-f002:**
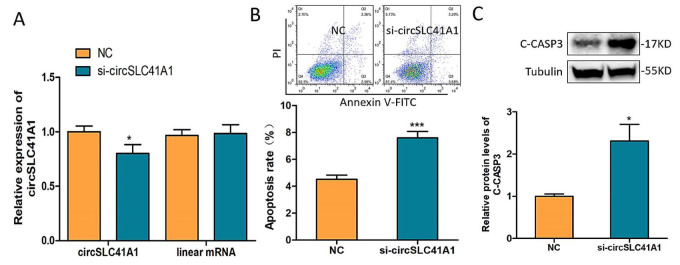
circSLC41A1 inhibits GCs apoptosis. (**A**) Expression of circSLC41A1 and its corresponding linear mRNA in GCs treated with siRNA control (NC) or si-circSLC41A1 RNA detected by qRT-PCR. (**B**) The apoptosis rate of GCs after circSLC41A1 knockdown examined by Annexin V-FITC/PI staining assay using flow cytometry at the top of panel B. (**C**) Protein levels of cleaved CASP3 after circSLC41A1 knockdown were analyzed by western blot. Data are expressed as the mean ± SEM of three experiments; * *p* < 0.05, *** *p* < 0.001.

**Figure 3 ijms-23-01509-f003:**
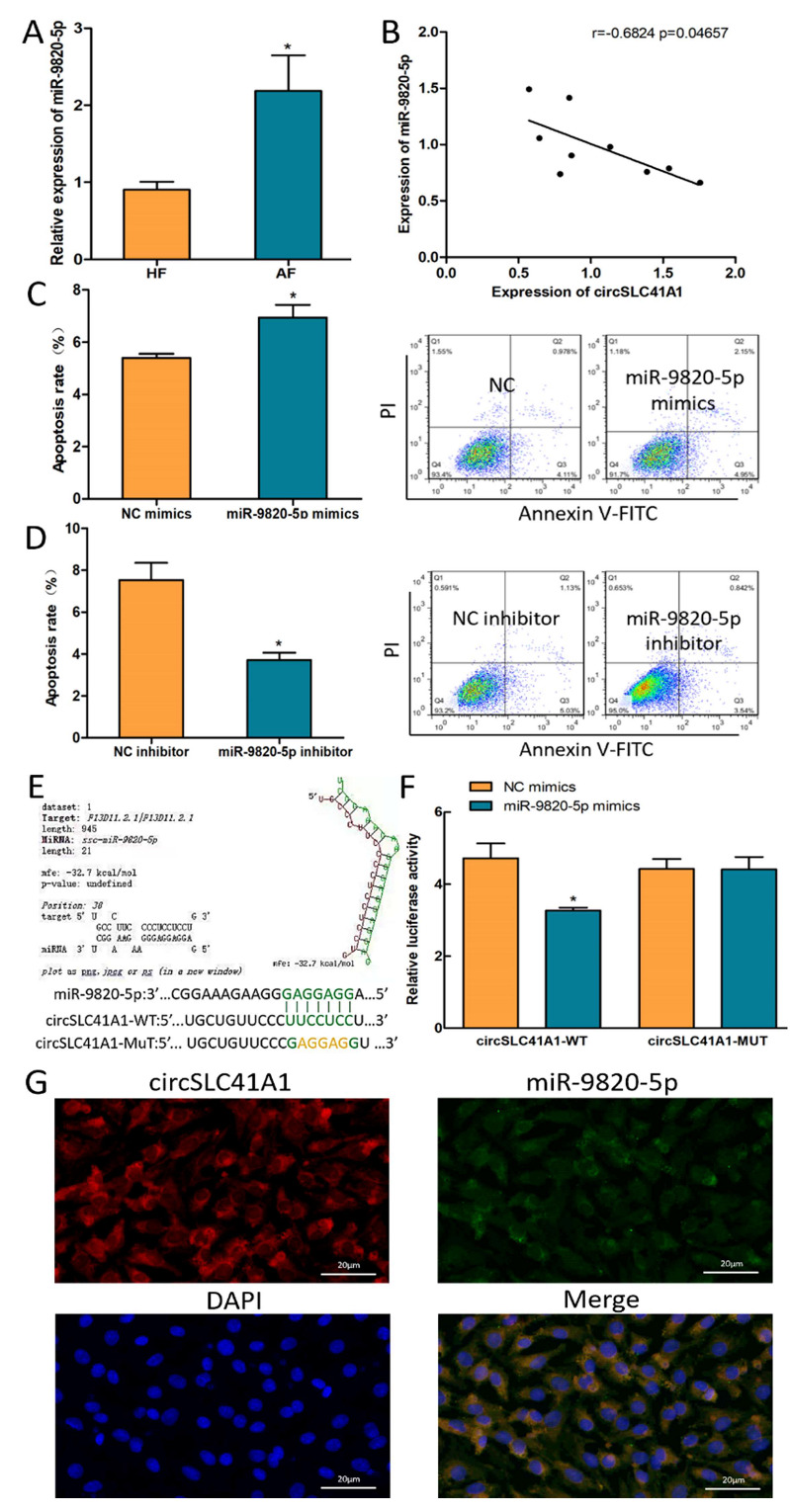
circSLC41A1 is a possible sponge for miR-9820-5p. (**A**) Differentially expressed miR-9820-5p in HFs and AFs detected by qRT-PCR. (**B**) Negative correlation between circSLC41A1 and miR-9820-5p expression levels in individual follicles (*n* = 9). (**C**,**D**) Shift of GC apoptosis rates after miR-9820-5p mimics and inhibitor transfection detected by flow cytometry. (**E**,**F**) The binding of miR-9820-5p of circSLC41A1 was verified by dual-luciferase activity analysis. (**G**) The subcellular co-localization of circSLC41A1 (labeled by red fluorescence) and miR-9820-5p (labeled by green fluorescence) in GCs verified by FISH. The nuclei were stained by DAPI (blue). Scale bar: 20 μm. Data are expressed as the mean ± SEM of three experiments; * *p* < 0.05.

**Figure 4 ijms-23-01509-f004:**
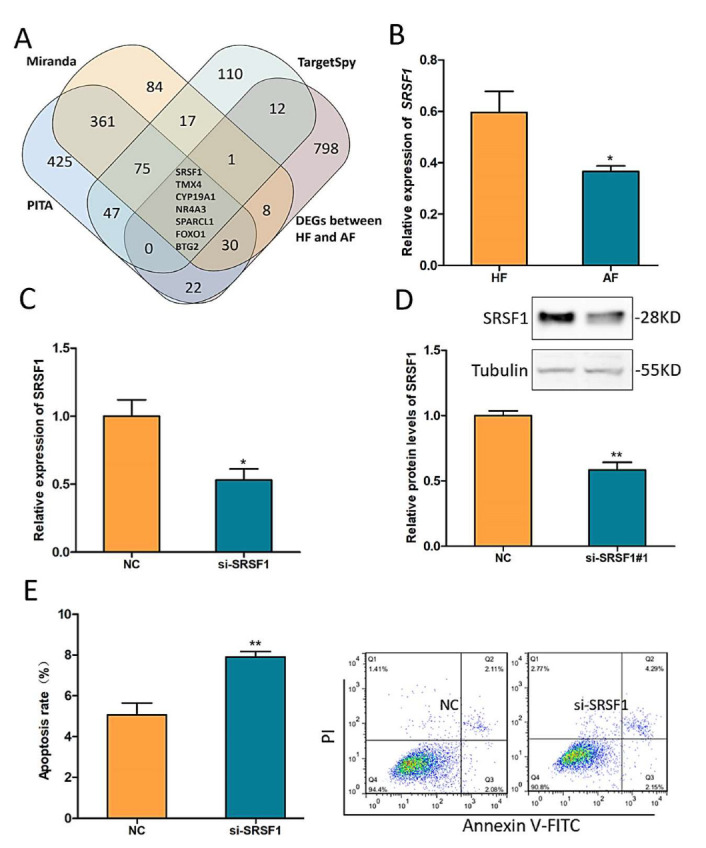
*SRSF1* resists GC apoptosis and follicular atresia. (**A**) Venn diagram of the DEGs (differentially expressed genes) between HFs and AFs and miR-9820-5p target genes predicted by PITA, Miranda, and TargetSpy. (**B**) Differentially expressed *SRSF1* in HFs and AFs detected by qRT-PCR. (**C**,**D**) Effective knockdown of *SRSF1* by si-SRSF1 qRT-PCR and western blotting (WB). (**E**) The shift of GC apoptosis rates after *SRSF1* knockdown was detected by flow cytometry. Data are expressed as the mean ± SEM of three experiments; * *p* < 0.05, ** *p* < 0.01.

**Figure 5 ijms-23-01509-f005:**
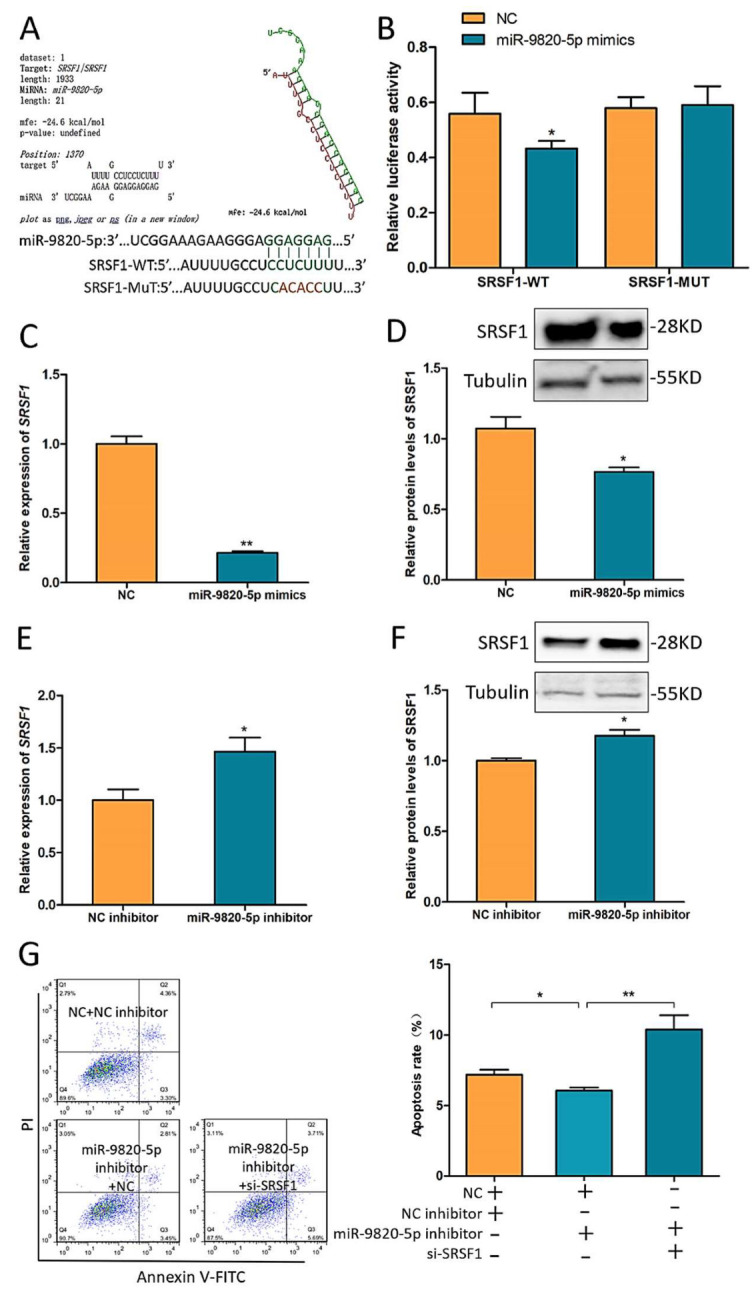
miR-9820-5p promotes pGCs apoptosis by targeting *SRSF1*. (**A**,**B**) The binding of miR-9820-5p and SRSF1-WT was predicted by RNAhybrid and verified by dual-luciferase activity analysis. (**C**,**D**) *SRSF1* mRNA and protein levels after transfection of miR-9820-5p mimics. (**E**,**F**) *SRSF1* mRNA and protein levels after transfection of miR-9820-5p inhibitors. (**G**) The shift of GC apoptosis rates after co-transfection of miR-9820-5p inhibitors and si-SRSF1 detected by flow cytometry. Data are expressed as the mean ± SEM of three experiments; * *p* < 0.05, ** *p* < 0.01.

**Figure 6 ijms-23-01509-f006:**
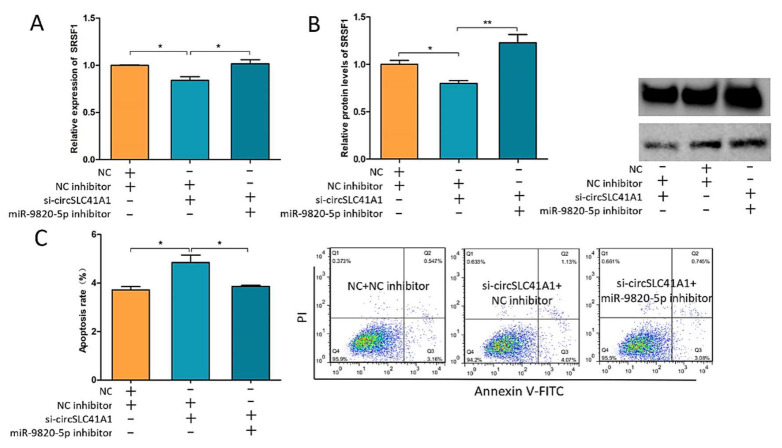
circSLC41A1 inhibits GCs apoptosis by upregulating *SRSF1* through sponging miR-9820-5p. (**A**) The shift of *SRSF1* mRNA levels after co-transfection of si-circSLC41A1 and miR-9820-5p inhibitors detected by qRT-PCR. (**B**) The shift of *SRSF1* protein levels after co-transfection of the si-circSLC41A1 and miR-9820-5p inhibitors detected by WB. (**C**) Change of GC apoptosis rates after co-transfection of si-circSLC41A1 and miR-9820-5p inhibitors detected by flow cytometry. Data are expressed as the mean ± SEM of three experiments; * *p* < 0.05, ** *p* < 0.01.

**Figure 7 ijms-23-01509-f007:**
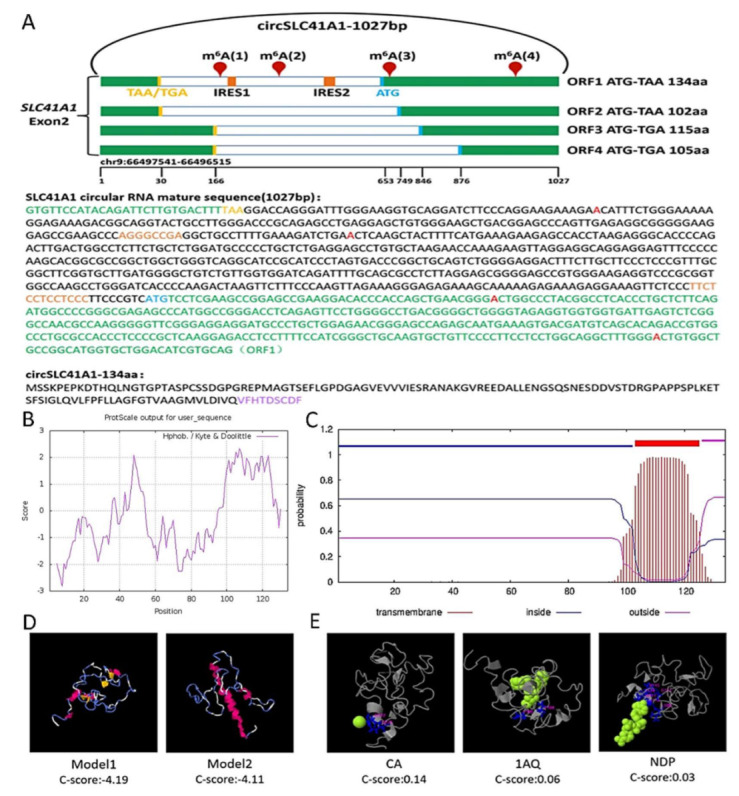
Evaluation of the coding ability of circSLC41A1. (**A**) Upper panel: the putative open reading frame (ORF) in circSLC41A1. Note that the circSLC41A1 junction is inside the ORF1. Lower panel: the sequences of the putative ORF are shown in green, internal ribosomal entrance site (IRES) sequences are shown in orange, the potential m^6^A site is shown in red, and specific amino acid sequences of ORF1 are shown in purple. (**B**) Hydrophobicity analysis of circSLC41A1-134aa. The ordinate score greater than 0 indicates hydrophobicity and less than 0 indicates hydrophilicity. (**C**) The transmembrane domain prediction of circSLC41A1-134aa. The transmembrane domain is shown in red, the intramembranous region is shown in blue, and the extramembranous region is shown in fuchsia. (**D**) The tertiary structure prediction of circSLC41A1-134aa. α-helix, β-sheet, and a random coil are shown in red, blue, and yellow, respectively. (**E**) The ligand prediction of 134aa. The green segments represent ligands and red segments represent the binding domains.

**Figure 8 ijms-23-01509-f008:**
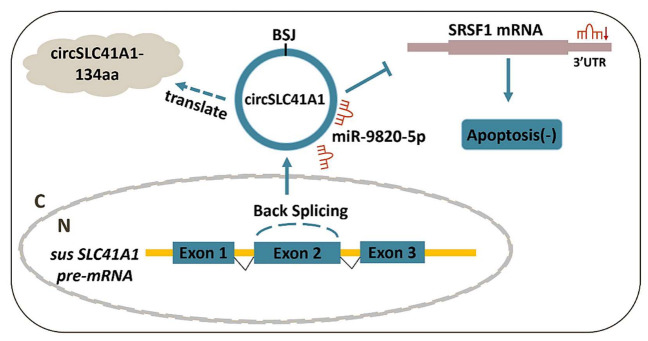
Schematic diagram of circSLC41A1 functions in GC apoptosis. circSLC41A1 can function by competing with *SRSF1* mRNA for miR-9820-5p interactions, thereby alleviating the post-transcriptional repression of *SRSF1* and thus resisting GC apoptosis and follicular atresia. circSLC41A1 also had the potential to encode small circSLC41A1-134aa peptides.

## Data Availability

The datasets used during the current study are available from the corresponding author on reasonable request.

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
