# Peer review of "circSLC41A1 Resists Porcine Granulosa Cell Apoptosis and Follicular Atresia by Promoting SRSF1 through miR-9820-5p Sponging"

_ijms, 2022, doi:10.3390/ijms23031509_

Round 1

Reviewer 1 Report

The publication titled "circSLC41A1 resists porcine granulosa cell apoptosis and ..." by Huiming Wang et al is a pretty good job. All in all, it is also quite ambitious and worth publishing. However, in my opinion, it has a lot of errors - they are not factual errors, but a lot of such shortcomings that should be considered before its publication, unfortunately, it has.
However, I would like to point out that these are not things that capture her anything because the substantive side of this publication is really strong. But the presentation of the results and some of the wording leave something to be desired. Below are my comments.
1. The title sounds somehow strange in English, maybe it is worth considering a slight revision of it.
2. the abstract, especially it's first part, is too general - it can be delicately rewritten. Such a division in the abstract into (1) Background, (2) Methods, etc. I do not know if it is needed. If I am interested in it, I will simply look at the M&M section and find out what the authors used in terms of methods, etc. Strange - I propose to change it.
3. Keywords - should be more general - so far (at least some of them) narrow this work terribly.
4. The introduction is fun. I would add a little more citations and I would elaborate the last paragraph here rather than with just one sentence about the purpose of the work.
5. Figure 2 - you can think to enlarge it a bit, for now, it is slightly illegible - especially its top.
6. Figure 5 panels A and G - need to be more elaborated because so far nothing can be seen in them.
7. Same Figure 6 Panel C.
8. In figure 7 - what are the panels B, C, and D for - I don't understand - such artificial enlargement of the work - cannot be given to e.g. SM? Mercy.
9. Discussion is by far the weakest part of this job. It has not started well yet and is now over. In fact, it is not known how it differs from the description of the results. This part needs to be definitely improved, and practically all of it.
10. Summary - or the lack of it, I advise you to think about it and develop it further - the subject of the work itself is extremely interesting. It is a pity that the further you read this manuscript, the weaker it gets. It's a pity because the job is generally good.
But because of this approach, it seems to me that other Reviewers may not be so kind.
To sum up, please pay attention to the corrections because the work is fun and the topic itself is ambitious. And just a few more steps are needed to publish this work. I recommend some decent fixes for now.

Author Response

Dear reviewer,

We appreciate the recognition and the comments, and we are grateful that we improved our manuscript according to the comments. Please find all the changes listed point to point below. Please refer to the attachment for specific modifications. Our responses are in red.

1. The title sounds somehow strange in English, maybe it is worth considering a slight revision of it.

Response 1: We agree with you. It is a bit tricky to put all the necessary information in one title. Here are two alternative titles below and we would like to hear your opinion.

A-circSLC41A1 promotes SRSF1 through miR-9820-5p sponging to resist porcine granulosa cell apoptosis and follicular atresia

B-circSLC41A1 promotes SRSF1 as a miR-9820-5p sponge to resist porcine granulosa cell apoptosis and follicular atresia

2. the abstract, especially it's first part, is too general - it can be delicately rewritten. Such a division in the abstract into (1) Background, (2) Methods, etc. I do not know if it is needed. If I am interested in it, I will simply look at the M&M section and find out what the authors used in terms of methods, etc. Strange - I propose to change it.

Response 2: We appreciate the suggestion and we have removed all the small titles in the abstract and tightened the language to focus on the main ideas. The journal requires to contain a” brief description of the main methods or treatments applied”, thus we tried our best to simplify this part. The new abstract is as below:

Abstract:

Ovarian granulosa cell (GC) apoptosis is the major cause of follicular atresia. Regulation of non-coding RNAs (ncRNA) was proved to be involved in regulatory mechanisms of GC apoptosis. circRNAs have been recognized to play important roles in cellular activity. However, the regulatory network of circRNAs in follicular atresia has not been fully validated. In this study, we reported a new circRNA, circSLC41A1, which expressed higher in healthy than atretic follicles and confirmed its circular structure using RNase R treatment. The resistant function of circSLC41A1 during GC apoptosis was detected by si-RNA transfection, and the competitive binding of miR-9820-5p by circSLC41A1 and SRSF1 using the dual-luciferase reporter assay and co-transfection of their inhibitor or siRNA. Additionally, we predicted the protein-coding potential of circSLC41A1 and analyzed the structure of circSLC41A1-134aa. Our study illustrated that circSLC41A1 enhanced SRSF1 expression through competitive binding of miR-9820-5p, demonstrated a circSLC41A1/miR-9820-5p/SRSF1 regulatory axis in follicular GC apoptosis. The study added the knowledge of posttranscriptional regulation of follicular atresia, and provided insight into the protein-coding function of circRNA.

3. Keywords - should be more general - so far (at least some of them) narrow this work terribly.

Response 3: We couldn’t agree more. We have added ‘circRNA’, ‘non-coding RNA’ and ‘ ceRNA’ in the keywords.

4. The introduction is fun. I would add a little more citations and I would elaborate the last paragraph here rather than with just one sentence about the purpose of the work.

Response 4: We have added 4 citations regarding circRNA studies in porcine, bovine, and goat ovarian follicles in Line 51-56. And we also added the research significance of this work in the last paragraph of the introduction.

5. Figure 2 - you can think to enlarge it a bit, for now, it is slightly illegible - especially its top.

Response 5: We have examined the figures, and we wonder if you mean figure 3 rather than figure 2? Because figure 2 should be clear enough, figure 3 was indeed too dense. Thus, we remade figure3 using a vertical layout to enlarge each figure of the set. We believe it looks much better now. Please let us know if we misunderstood this point.

6. Figure 5 panels A and G - need to be more elaborated because so far nothing can be seen in them.

Response 6: We have remade figure 5 using a vertical layout and added descriptions about Figure 5 panels A and G.

7. Same Figure 6 Panel C.

Response 7: We have added descriptions about Figure 6C.

8. In figure 7 - what are the panels B, C, and D for - I don't understand - such artificial enlargement of the work - cannot be given to e.g. SM? Mercy.

Response 8: We especially appreciate this point and realized our roughness in the 2.7 section. We have taken a couple of steps to improve this part. First, we added several new results to complete the prediction of circSLC41A1-134aa biochemical properties, such as its hydrophily and transmembrane domain in Line 223-224. Second, we have rearranged Figure 7 and kept more informative results in the manuscript. The less important ones such as the homology analysis were arranged in supplementary figure 2. Thirdly, we have revised the figure legend accordingly.

9. Discussion is by far the weakest part of this job. It has not started well yet and is now over. In fact, it is not known how it differs from the description of the results. This part needs to be definitely improved, and practically all of it.

Response 9: We have largely rewritten the discussion part, especially Line 271-315. We tried to discuss in depth the possible signaling pathways downstream of SRSF1 in cell apoptosis and, thanks to your 8th comment above, we also discussed two possible ways through which circSLC41 may affect granulosa cell function and survival.

10. Summary - or the lack of it, I advise you to think about it and develop it further - the subject of the work itself is extremely interesting. It is a pity that the further you read this manuscript, the weaker it gets. It's a pity because the job is generally good.

Response 10: We have updated a more comprehensive summary with the conclusion and prospect in Line 309-315.

Best regards

Your sincerely

Huiming Wang

Nanjing Agricultural University

2019105085@njau.edu.cn

Reviewer 2 Report

To investigate regulatory mechanisms by ncRNA in the ovarian granulosa cell apoptosis, the authors examined circSLC41A1, coded by SLC41A1, resisted GC apoptosis and follicular atresia by enhancing the expression of the SR protein splicing factor SRSF1 through competitive binding of miR-9820-5p. Their results showed circSLC41A1 promotes SRSF1 expression by sponging miR-9820-5p and inhibits GC apoptosis and follicular atresia and demonstrated a circSLC41A1/miR-9820-5p/SRSF1 regulatory axis in follicular GC apoptosis. This article was well written and could be acceptable for publication. 

Author Response

Dear reviewer:

Thank you very much for your suggestions and recognition. We have further improved the abstract, introduction, and discussion of the manuscript, and supplemented the content of result 7. Please see the attachment for specific revisions. If you have more suggestions, please discuss with us and look forward to your reply.

Best regards

Yours sincerely

Huiming Wang

Nanjing Agricultural University

2019105085@njau.edu.cn

Round 2

Reviewer 1 Report

The authors improved the work very well, which was quite good anyway.
Congratulations and I wish you continued success.